# Generation Scotland: an update on Scotland's longitudinal family health study

Hannah Milbourn ©,[1,2] Daniel McCartney ©,[1] Anne Richmond,[1]
Archie Campbell,[1] Robin Flaig,[2] Sarah Robertson,[1,2] Chloe Fawns-Ritchie,[3,4]
Caroline Hayward,[1] Riccardo E Marioni ©,[1] Andrew M McIntosh ©,[1,5]
David J Porteous,[1] Heather C Whalley ©,[1,5,6] Cathie Sudlow ©[6]

For numbered affiliations see end of article.

**Correspondence to**
Professor Heather C Whalley; heather.whalley@ed.ac.uk

## ABSTRACT

**Purpose** Generation Scotland (GS) is a large family-based cohort study established as a longitudinal resource for research into the genetic, lifestyle and environmental determinants of physical and mental health. It comprises extensive genetic, sociodemographic and clinical data from volunteers in Scotland.

**Participants** A total of 24 084 adult participants, including 5501 families, were recruited between 2006 and 2011. Within the cohort, 59% (approximately 14 209) are women, with an average age at recruitment of 49 years. Participants completed a health questionnaire and attended an in-person clinic visit, where detailed baseline data were collected on lifestyle information, cognitive function, personality traits and mental and physical health. Genotype array data are available for 20 026 (83%) participants, and blood-based DNA methylation (DNAm) data for 18 869 (78%) participants. Linkage to routine National Health Service datasets has been possible for 93% (n=22 402) of the cohort, creating a longitudinal resource that includes primary care, hospital attendance, prescription and mortality records. Multimodal brain imaging is available in 1069 individuals.

**Findings to date** GS has been widely used by researchers across the world to study the genetic and environmental basis of common complex diseases. Over 350 peer-reviewed papers have been published using GS data, contributing to research areas such as ageing, cancer, cardiovascular disease and mental health. Recontact studies have built on the GS cohort to collect additional prospective data to study chronic pain, major depressive disorder and COVID-19.

**Future plans** To create a larger, richer, longitudinal resource, 'Next Generation Scotland' launched in May 2022 to expand the existing cohort by a target of 20 000 additional volunteers, now including anyone aged 12+ years. New participants complete online consent and questionnaires and provide postal saliva samples, from which genotype and salivary DNAm array data will be generated. The latest cohort information and how to access data can be found on the GS website (www. generationscotland.org).

### STRENGTHS AND LIMITATIONS OF THIS STUDY

⇒ The Generation Scotland cohort has a wide range of demographic, lifestyle, health and genetic data and includes participants from a wide range of sociodemographic backgrounds.

⇒ Linkage to a variety of longstanding, routine National Health Service (NHS) datasets, which are particularly rich and diverse in Scotland, creates a breadth of longitudinal phenotype information.

⇒ Planned linkages to medical images and radiology reports from across the NHS in Scotland will enrich the linked health data, while additional linkages to administrative data (eg, education records) will broaden the information on participants and support research on the wider determinants of health and well-being.

⇒ The cohort is relatively small, by contemporary standards for population-based cohorts. However, this issue can often be addressed through joint analyses with other population-based cohorts and participation in genetic data consortia.

## INTRODUCTION

Generation Scotland (GS) is a longitudinal health study established as a family-based and population-based resource for the study of the genetic, lifestyle and environmental determinants of common complex diseases. Non-communicable diseases, such as cancer, diabetes, stroke, heart, liver and lung disease, are the leading cause of morbidity and mortality in Scotland.[1 2] The majority of common health disorders of public health concern are a result of a complex interaction between genes and environment. The GS cohort is rich in genetic and phenotypic information through data collection, sample assays and linkage to routine electronic health records. As a bio-resource for medical research, GS aims to support research to establish the determinants of physical and mental health and improve the prevention, diagnosis and treatment of common diseases.

GS was founded as a multi-institutional, cross-disciplinary collaboration between

the Universities of Aberdeen, Dundee, Edinburgh and Glasgow and the National Health Service (NHS) Scotland, with key resources, expertise and input from the Medical Research Council Human Genetics Unit, the National eScience Centre and the Scottish School of Primary Care. A list of current staff working on developing and maintaining the GS resource and members of the scientific steering committee is provided in online supplemental appendix A.

Between 2006 and 2011, over 24 000 adult volunteer participants completed questionnaires, attended a clinic visit or participated by post, and consented to genetic studies, linkage to their medical records and to be recontacted for future research.[3]

In 2022, GS launched Next Generation Scotland (NGS), aiming to expand the existing cohort by recruiting 20 000 new participants, newly including 12–17 year olds, meeting an unmet need to study adolescent health. Data are collected via an online questionnaire and saliva sample collection by post.

An earlier cohort profile paper described the baseline recruitment.[4] Here, we report data enrichment of the cohort, including new biological data, longitudinal data linkage and recontact studies. We highlight the extent and nature of the data now available to researchers, summarise the use and impact of GS since commencement in 2006 and outline the current and future plans for NGS.

## COHORT DESCRIPTION

## PARTICIPANT RECRUITMENT

The original GS:Scottish Family Health Study (SFHS) protocol and baseline data profile have been described previously.[3 4] Briefly, potential participants aged 35–65 years (study probands) were selected from lists of collaborating general medical practices in Scotland. They were invited to participate in the study and asked to identify at least one adult (18+ years old) first-degree relative to invite to the study. This included volunteers from the Glasgow and Tayside areas from 2006 to 2011 and was extended to include Ayrshire, Arran and Grampian in 2010. Participants completed a Pre-Clinic Questionnaire (PCQ) before attending a research clinic in Glasgow, Dundee, Perth, Aberdeen or Kilmarnock. In total, 126 000 individuals were invited to participate, of whom 6665 responded and met the study criteria (response rate of 5.3%). An additional 17 419 family members were recruited via these probands. The original GS:SFHS cohort therefore consists of 24 084 participants.

## Baseline data collection

All 24 084 participants completed a PCQ collecting a range of demographic, social characteristics, personal behaviours and self-reported health data. Information collected included smoking status, alcohol consumption

and personal and family disease history. Information was also collected on the birthplaces, by local council area, of participants and their parents and grandparents born in Scotland. In 2009, revisions were made to several questions within the PCQ for machine readability; the period prior to this was termed phase 1 (n=9967) and the period thereafter was termed phase 2 (n=14 117).

Most participants (21 476) also attended a research clinic, where physical measurements included height, weight, heart rate, systolic and diastolic blood pressure, ECG and body composition analysis. Standardised and well-validated assessments of cognitive function, personality and mental health included the 28-item General Health Questionnaire, Eysenck Personality Questionnaire, Structured Clinical Interview for DSM Disorders, Mood Disorder Questionnaire, Schizotypal Personality Questionnaire, Mill Hill Vocabulary test and WAIS-III logical memory test. All baseline measures collected are reported in the previous cohort profile.[4]

## Cohort characteristics at recruitment

Of the 24 084 participants within the original GS:SFHS cohort, approximately 14 209 (59%) are female, with an average age at recruitment of 49 years. In total, 87% (approximately 20 953) of participants were born in Scotland and 97% (approximately 23 361) born within the UK. The cohort includes a range of sociodemographic characteristics, although compared with the Scottish population participants have a higher education level and lower deprivation index (table 1).

Family groups of at least two first-degree relatives were identified and assigned a shared family identity number. Pedigrees were constructed using relationship information provided by study participants and validated with genetic kinship information following genotyping. The cohort contained 1361 singletons (with no relatives in the study) and 5501 families of at least 2 people, with a mean size of 4.1 family members.

## Longitudinal data linkage

Linkage to extensive and longstanding NHS Scotland records, both retrospective and prospective, creates a longitudinal cohort from baseline. The linkage data available within GS have not previously been described. At the time of writing, participants have up to 16-year follow-up data since recruitment in 2006. Routine NHS data are obtained through collaboration with the Health Informatics Centre at the University of Dundee, with linkage performed using the Community Health Index (CHI) number. CHI numbers are used across NHS Scotland services and are unique to each general practice (GP)-registered individual living in Scotland. In total, 93% of GS participants consented to linkage and had a CHI number available. For individuals with CHI linkage, 89% also have genome-wide genotype data available (see Laboratory samples and molecular assays below).

Table 2 and figure 1 show the range of datasets linked to the GS cohort, their periods of coverage and the

**Table 1** Demographic and lifestyle characteristics of GS:SFHS cohort (n=24 084) and comparison to the Scottish population

| Characteristic | N | % (or median and IQR) | Scotland average |
|---|---|---|---|
| Sex | | | |
| Female | 14 157 | 58.8% | 51.5%* |
| Male | 9927 | 41.2% | 48.5%* |
| Age | | | |
| Median age (IQR) | 24 084 | 49 (36–59) | 41.3* |
| Ethnicity | | | |
| White | 22 826 | 94.8% | 96.0%* |
| Other | 262 | 1.1% | 4.0%* |
| Missing | 996 | 4.1% | |
| SIMD quintiles | | | |
| 5 (least deprived) | 6571 | 27.3% | 20.6%† |
| 4 | 5419 | 22.5% | 21.1%† |
| 3 | 3437 | 14.3% | 19.8%† |
| 2 | 2987 | 12.4% | 19.3%† |
| 1 (most deprived) | 2724 | 11.3% | 19.2%† |
| Missing | 2946 | 12.2% | |
| Rural Urban Classification codes | | | |
| 1—large urban areas | 7369 | 30.6% | 37.8%‡ |
| 2 | 6679 | 27.7% | 33.9%‡ |
| 3 | 2032 | 8.4% | 8.6%‡ |
| 4 | 1006 | 4.2% | 2.6%‡ |
| 5 | 2662 | 11.1% | 11.6%‡ |
| 6—remote rural areas | 1390 | 5.8% | 5.5%‡ |
| Missing | 2946 | 12.2% | |
| Employment status (up to 75 years) | | | |
| Employed (full time or part-time) | 14 808 | 64.6% | 73.2%§ |
| Unemployed | 970 | 4.2% | 3.90%§ |
| Retired | 3080 | 13.4% | |
| Other | 1808 | 7.9% | |
| Missing | 2255 | 9.8% | |
| Education—highest qualification attained | | | |
| No qualification | 3145 | 13.3% | 26.8%* |
| Lower secondary school | 3611 | 15.3% | 23.1%* |
| Higher secondary school | 2452 | 10.4% | 14.3%* |
| College level | 6450 | 27.3% | 9.7%* |
| University level | 7330 | 31.0% | 26.1%* |
| Other | 624 | 2.6% | |
| Missing | 1738 | 7.2% | |
| Smoking status | | | |
| Current smoker | 3997 | 16.6% | 11.3%¶ |
| Ex-smoker | 6964 | 28.9% | 23.2%¶ |

Continued

**Table 1** Continued

| Characteristic | N | % (or median and IQR) | Scotland average |
|---|---|---|---|
| Non-smoker | 12 227 | 50.8% | 65.6%¶ |
| Missing | 896 | 3.7% | |
| Alcohol consumption | | | |
| Median alcohol units per week (IQR) | 21 737 | 8 (2–15) | 6.1¶ |

*Based on 2011 Scottish Census.[38–40]
†Based on National Records of Scotland (NRS) Population Estimates by SIMD.[41]
‡Based on NRS Scottish Government Urban Rural Classification 2020.[42]
§Based on the ONS Annual Population Survey 2021.[43] Individuals aged 18–64 years.
¶Based on The Scottish Health Survey 2021.[1]
GS, Generation Scotland; SFHS, Scottish Family Health Study; SIMD, Scottish Index of Multiple Deprivation.

numbers of participants with linked data available for each. Additional details are provided in online supplemental appendix B. Beyond linkages to hospital episodes, primary care, cancer and death registries and community electronic prescribing, GS has linkage to a range of other datasets via participants' CHI numbers, including routine laboratory tests, dental data (from the Management Information & Dental Accounting System) and the Scottish Drug Misuse Database, offering unique phenotype information distinct from other population-based cohort research resources. COVID-19 testing, diagnoses and vaccination records are also available for the period of 2020–2022.

Regular data refreshes are received, and new datasets are added to enhance and continue the follow-up of participants over time. Planned additional linkages include incorporating NHS Scotland routine NHS radiology images, including X-rays, CT and MRI scans (Scottish Medical Imaging), imaging reports and retinal scans, which will provide new research opportunities not available in other population-based cohorts. Text-based radiology report linkage has already been applied to a study of stroke phenotyping in GS participants.[5]

### Cohort morbidities
Participant self-reported disease prevalence (at recruitment) is shown in table 3 alongside longitudinal data on morbidities obtained through data linkage to primary care (GP) data, Scottish Morbidity Records (SMR) and National Records of Scotland death records. ICD and Read Codes to define disease prevalence were derived from Gadd et al[6] using CALIBER code lists, detailed in full in online supplemental appendix table C–E. Data are available up to 2020 for GP records, cancer registries and 2022 for hospital admissions (SMR01) and mortality records. Diagnoses of 3006 hypertension, 2197 asthma, 2371 depression, 2558 osteoarthritis and 1701 heart disease cases are reported across all primary and secondary care and mortality linked data sources.

**Table 2** Summary of linked data sources

| | Total participants | Genome-wide genotype data available | |
| --- | --- | --- | --- |
| | | N | % |
| CHI linkage | 22 403 | 19 960 | 89.1% |
| Scottish Morbidity Records | | | |
| Outpatient Attendance (SMR00) | 21 271 | 19 159 | 90.1% |
| General/Acute Inpatient and Day Case (SMR01) | 18 249 | 16 467 | 90.2% |
| Maternity Inpatient and Day Case (SMR02) | 8239 | 7537 | 91.5% |
| Mental Health Inpatient and Day Case (SMR04) | 578 | 516 | 89.3% |
| Scottish Cancer Registry (SMR06) | 3606 | 3207 | 88.9% |
| Scottish Birth Record (SMR11) | 3246 | 2864 | 88.2% |
| Primary care | | | |
| General practice (GP) | 19 676 | 17 823 | 90.6% |
| GP out of hours | 8533 | 7700 | 90.2% |
| NHS24 | 12 326 | 11 108 | 90.1% |
| Accident and emergency | 15 249 | 13 778 | 90.4% |
| Other datasets | | | |
| Routine laboratory test results | 19 090 | 17 521 | 91.8% |
| ICU episode data (SICSAG) | 361 | 324 | 89.8% |
| Deaths (NRS deaths data) | 1659 | 1376 | 82.9% |
| Diabetes registry (SCI-DC) | 1423 | 1241 | 87.2% |
| Prescription dispensing (PIS) | 21 486 | 19 347 | 90.0% |
| COVID-19 vaccinations | 19 128 | 17 358 | 90.7% |
| COVID-19 testing (ECOSS) | 16 537 | 14 995 | 90.7% |
| Dental (MIDAS) | 19 871 | 17 904 | 90.1% |
| Scottish drug misuse database (SDMD) | 76 | 63 | 82.9% |

Total number of participants and proportion of individuals with genome-wide genotyping data available for all linked data.
CHI, Community Health Index; ECOSS, Electronic Communication of Surveillance in Scotland; MIDAS, Management Information & Dental Accounting System; NHS24, Scottish national telehealth and telecare organisation; NRS, National Records of Scotland; PIS, Prescribing Information System; SCI-DC, Scottish Care Information—Diabetes Collaboration; SICSAG, Scottish Intensive Care Society Audit Group.

As an extended example, figure 2 shows the proportion of diabetes cases captured in secondary care and deaths records as described above and enhanced with additional data sources available within GS (The Scottish Care Information—Diabetes Collaboration (SCI-DC), the Prescribing Information System (PIS) and routine laboratory testing data). A total of 1861 diabetes (types 1 and 2) cases were recorded in at least one source, cohort prevalence of 7.7%. The SCI-DC captures 74% of all recorded cases. Prescriptions of metformin hydrochloride or insulin within the PIS captured 73% of diabetes cases. Linked routine laboratory testing data contained results for any glycated haemoglobin (HbA1c) tests conducted as an indication of average blood sugar levels. Individuals with percentage HbA1c in blood (HbA1c levels) above 6.5 (48 mmol/mol) were classified as diabetic (28% of cases captured).[7] We note that lower proportions of cases within the self-reported source reflects that these were collected at baseline while other sources extend to 2020/2022. The use of a combination of data sources provides an opportunity to capture a range of cases and develop detailed phenotype definitions.

### Laboratory samples and molecular assays

Participants who attended a research clinic also provided biological samples (including blood and urine) for genotyping and other assays (n=23 979). Saliva was provided for DNA extraction by a subset of participants not attending a clinic (2608 sent a saliva sample by post) and was used for DNA extraction for an additional 984 participants from whom blood could not be obtained (total 3592). DNA was extracted from blood and saliva for 85% of participants (n=20 471). Basic biochemistry assays were performed on the baseline serum samples measuring creatinine, glucose, potassium, sodium, urea and cholesterol levels. Here, we provide an update on the genotyping methods conducted since baseline collection.

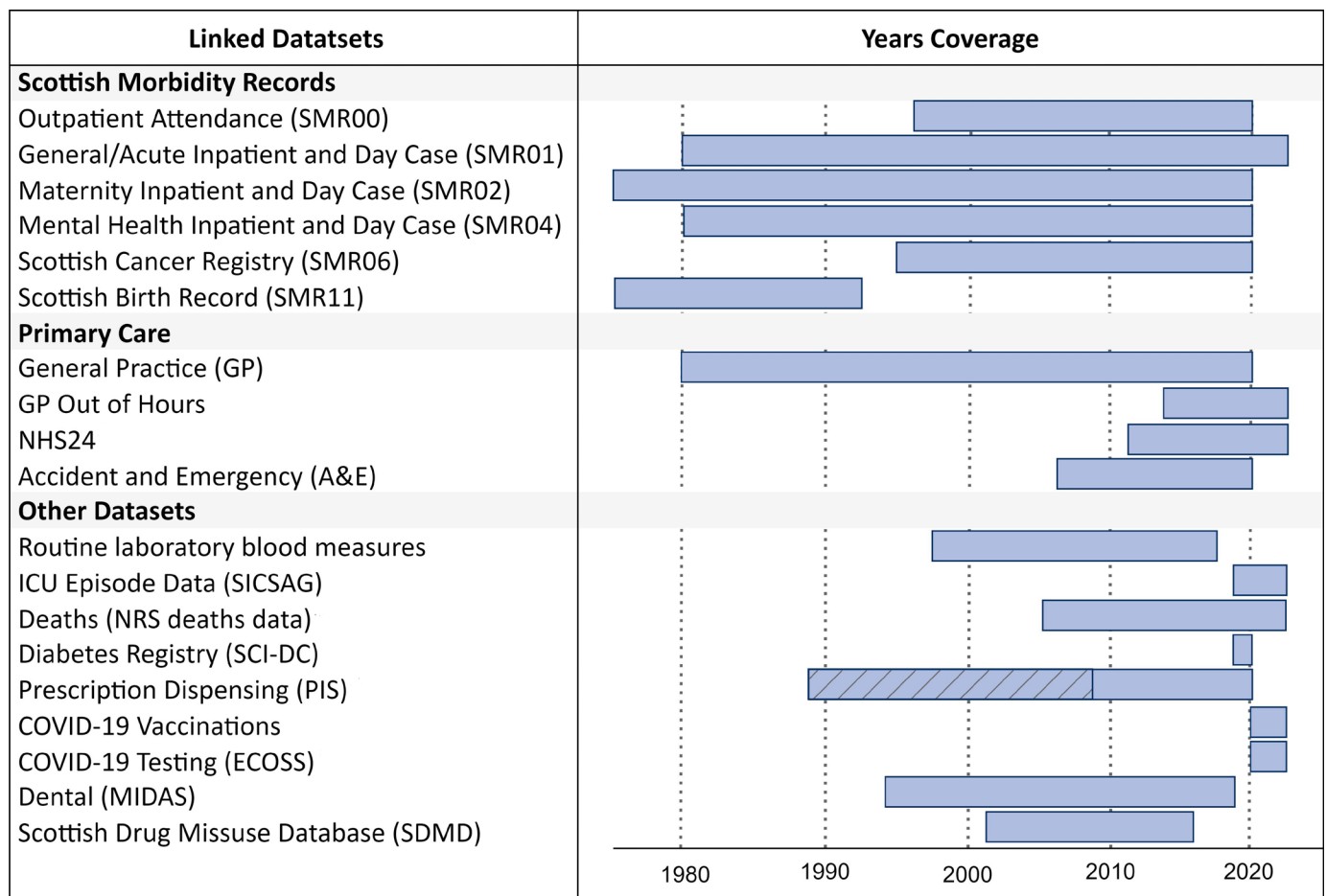

| Linked Datatsets | Years Coverage |
|---|---|

**Scottish Morbidity Records**
- Outpatient Attendance (SMR00)
- General/Acute Inpatient and Day Case (SMR01)
- Maternity Inpatient and Day Case (SMR02)
- Mental Health Inpatient and Day Case (SMR04)
- Scottish Cancer Registry (SMR06)
- Scottish Birth Record (SMR11)

**Primary Care**
- General Practice (GP)
- GP Out of Hours
- NHS24
- Accident and Emergency (A&E)

**Other Datasets**
- Routine laboratory blood measures
- ICU Episode Data (SICSAG)
- Deaths (NRS deaths data)
- Diabetes Registry (SCI-DC)
- Prescription Dispensing (PIS)
- COVID-19 Vaccinations
- COVID-19 Testing (ECOSS)
- Dental (MIDAS)
- Scottish Drug Missuse Database (SDMD)

1980  1990  2000  2010  2020

**Figure 1**  Summary of linked data sources periods of coverage. Linkage dataset coverage ranges from 1975 to 2022. Shaded portion represents incomplete coverage for data up to 2009. ECOSS, Electronic Communication of Surveillance in Scotland; MIDAS, Management Information & Dental Accounting System; NHS24, Scottish national telehealth and telecare organisation; NRS, National Records of Scotland; PIS, Prescribing Information System; SICSAG, Scottish Intensive Care Society Audit Group; SCI-DC, Scottish Care Information—Diabetes Collaboration.

## Genomics

Genome-wide genotyping data are available for 20 026 (83%) of the original GS:SFHS participants.[4] Samples were genotyped using the Illumina HumanOmniExpressExome8V.1-2_A and HumanOmniExpressExome–8V.1_A and the Beadstudio-Gencall V.3 genotype calling algorithm. Quality control measures were implemented, filtering out samples with a call rate of <98% and SNPs with a call rate of <98%, HWE of $<1\times10^{-6}$ and MAF of ≤1%, leaving 20 026 samples and 630 207 SNPs. Phasing of the genotyped SNPs was carried out using SHAPEIT V.2.[8]

Genetic profiles have been imputed using three different reference panels: 1000 Genomes,[9] Haplotype Reference Consortium[10] and Trans-Omics for Precision Medicine (table 4).[11] After imputation, further quality control procedures removed duplicate and monomorphic SNPs as well as those with an imputation quality score of <0.4.

## Methylomics

DNA methylation (DNAm) data have been generated using the Illumina HumanMethylationEPIC BeadChip array for 18 869 GS samples at >850 000 CpG sites, from blood collected at the baseline appointment (2006–2011). At the time of writing, this is the largest DNAm dataset from a single population-based cohort. These samples were processed in four batches between 2017 and 2021 and are referred to as set 1 (n=5087), set 2 (n=459), set 3 (n=4450) and set 4 (n=8873). A subsequent genome-wide DNAm measurement is also available for 880 individuals across set 2 (n=508) and set 3 (n=372), from additional blood collected between 2015 and 2019. The DNAm resource will be described in detail in a separate report. Briefly, quality control was carried out in R using the packages ShinyMethyl and WateRmelon. Probes with a bead count of less than three or a high detection p value (>0.05) in more than 5% of samples were removed. Outlier probes were also removed based on visual inspection of the log median intensity of the methylated versus unmethylated signal per array. Samples were removed where there were sex mismatches or where 1% or more of cytosine–guanine dinucleotides had a high detection p value (>0.05). A superset of 18 869 baseline samples has also been generated from the four individual sets,

**Table 3** Self-reported prevalence of common morbidities at baseline and morbidities in Generation Scotland participants using linked data up to 2022

| Disease outcome | Baseline self-reported morbidities 2006–2011 | | Morbidity diagnoses from NHS-linked data sources 1980–2022* | | | | |
|---|---|---|---|---|---|---|---|
| | N | % participants | Primary care (GP data) | General/Acute Inpatient and Day Case (SMR01) | NRS deaths Data | Scottish Cancer Registry (SMR06) | Total† |
| Hypertension | 3257 | 13.8% | 1669 | 2436 | 160 | | 3464 |
| Asthma | 2652 | 11.2% | 1719 | 1111 | 10 | | 2292 |
| Depression‡ | 2196 | 9.3% | 2139 | 334 | <10 | | 2346 |
| Osteoarthritis | 1783 | 7.5% | 1861 | 1639 | <10 | | 2817 |
| Heart disease | 935 | 3.9% | 960 | 1673 | 342 | | 1885 |
| Diabetes | 804 | 3.4% | 536 | 968 | 149 | | 1200 |
| Rheumatoid arthritis | 431 | 1.8% | 199 | 207 | 16 | | 322 |
| Stroke | 352 | 1.5% | 514 | 434 | 72 | | 753 |
| Breast cancer | 345 | 1.5% | 355 | 596 | 82 | 541 | 650 |
| COPD§ | 276 | 1.2% | 554 | 693 | 150 | | 952 |
| Bowel cancer | 142 | 0.6% | 155 | 245 | 39 | 213 | 298 |
| Prostate cancer | 104 | 0.4% | 164 | 271 | 53 | 252 | 323 |
| Lung cancer | 63 | 0.3% | 94 | 205 | 152 | 174 | 234 |
| Dementia¶ | 41 | 0.2% | 198 | 245 | 175 | | 356 |
| COVID-19 | | | <10 | 145 | 38 | | 162 |

ICD and Read Code lists are detailed in full in online supplemental appendix table C–E.
*Dataset date coverage: GP data (1980–2020), SMR01 (1980–2022), National Records of Scotland (NRS) deaths data (2007–2022), SMR06 (1996–2020), SMR04 (1980–2020).
†Deduplicated across data sources.
‡Phase 1 participants (n=9967) were asked if they have been diagnosed with depression and in phase 2 (n=14 117) if they had been diagnosed with severe depression.
§Chronic obstructive pulmonary disease (COPD) data for phase 2 only (n=14 117).
¶Participants were asked if they have been diagnosed with Alzheimer's disease.
GP, general practice; NHS, National Health Service.

comprising 831 733 CpGs that passed quality control in all sets.

### Proteomics and metabolomics

Protein levels have been quantified in plasma samples from 1065 participants using the 5k SOMAscan V.4 array from SomaLogic. Tandem mass spectrometry has been performed on a subset of 860 participants' blood samples for which peripheral blood mononuclear cells were available. Quantification of 54 urinary metabolite biomarkers in 2743 GS participants' samples has been conducted by Nightingale Health using nuclear magnetic resonance.

### Recontact studies

Participants provided broad consent permitting use of data and samples for 'future medical research into health, illness and medical treatment'. This included consent to be recontacted for new studies, which has led to additional data collections since recruitment, summarised in table 5. Data from recontact studies can be linked to GS data and are retained by GS to be made available for other researchers through the GS access process.

The Stratifying Resilience and Depression Longitudinally (STRADL) substudy recruited from the existing GS cohort

to subtype major depressive disorder (MDD), using detailed clinical, cognitive and brain imaging assessments. From 2015 to 2017, 9905 GS participants completed a remote depression-focused questionnaire (including psychological resilience, coping style and response to psychological distress) and a subset (n=1189) attended a face-to-face assessment to conduct cognitive testing, multimodal MRI of brain scans (n=1085) and further bio-sample collection.[12]

In 2016, the DOLORisk study enhanced GS to study neuropathic pain (NP). The study received responses to a survey regarding presence or absence of chronic pain and NP from 7238 of 20 221 members of the GS cohort invited to participate (35.8% response rate), with a follow-up repeat survey (n=5292 responses) after 18 months (table 5).[13]

GS is a member of the European Prevention of Alzheimer's Dementia Consortium, an interdisciplinary research initiative with partners across European organisations aiming to improve the understanding of the early stages of Alzheimer's disease.[14] In 2016, 53 GS participants attended a 'screening visit' for the collection of fasting blood samples and a brain scan (MRI) with follow-up visits after 6 months, 1, 2, 3 and 4 years.

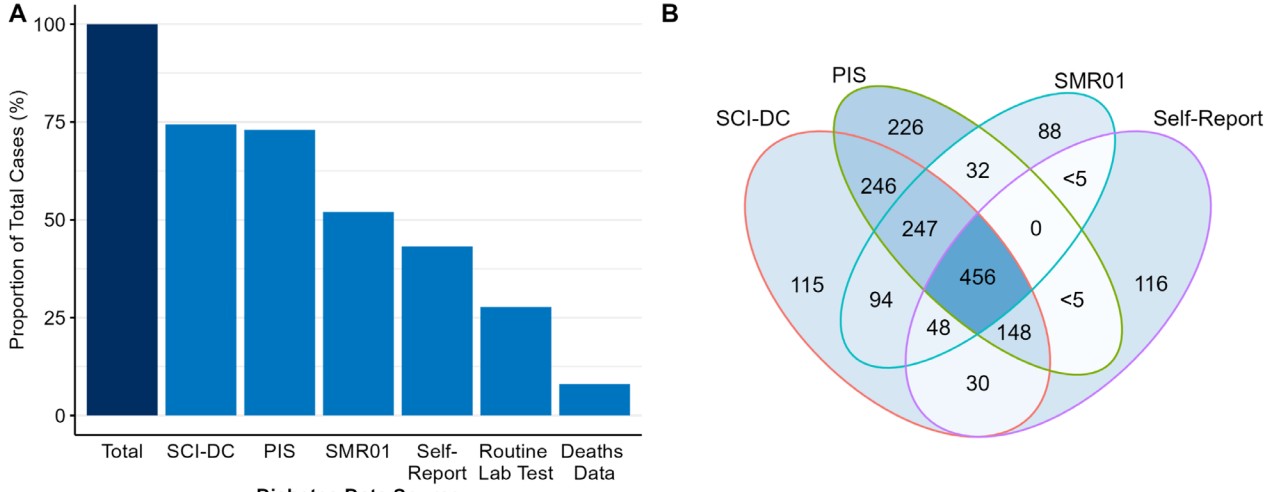

**Figure 2** (A). Proportion of diabetes cases captured in each data source (total cases n=1861). (B) Venn diagram of the concordance of cases among data sources (n=1852, excluding routine laboratory testing data for glycated haemoglobin (HbA1c) and deaths data). Self-report—prevalence reported by participant at baseline. Deaths data—National records of Scotland deaths data. SCI-DC, Scottish Care Information—Diabetes Collaboration; PIS, Prescribing Information System; SMR01, Scottish Morbidity Records: General/Acute Inpatient and Day Case; HbA1c, Routine laboratory testing data for haemoglobin A1c (level above 6.5%).

GS is partnering with Healthy AGeing in Scotland (HAGIS), a study of the health, economic and social circumstances of people over 50 years old in Scotland.[15] HAGIS is part of the Health & Retirement Study family of longitudinal ageing studies, which currently consists of longitudinal ageing studies in 16 countries around the world. GS recontacted 14 891 individuals in 2021–2022, with 2826 (19.0%) taking part in the HAGIS: COVID-19 Impact & Recovery Study.

Additional data collections conducted by the GS team include the COVIDLife surveys launched in April 2020 in response to the COVID-19 pandemic. The aim was to determine the impact of the pandemic on health and well-being. In total, 18 518 adult members of the UK public, including 4968 GS participants, participated in the surveys. Three COVIDLife surveys[16] and a Rural COVIDLife survey,[17] specific to rural Scottish volunteers, were conducted (total n=3365, GS participants n=712). In addition, three TeenCOVIDLife surveys,[18] for young people aged 12–18 years (n=7058), were run between April 2020 and June 2021. GS was part of the National Core Studies Longitudinal Health and Wellbeing programme established as part of the UK's pandemic response, including the coronavirus post-acute long-term effects: constructing an evidence base (CONVALESCENCE) long COVID study.[19] GS is also a participating cohort in COVIDMENT, a large-scale collaborative project between Northern European countries using data-rich population-based registry resources, biobanks and ongoing questionnaire data to further understanding of the mental health impact of the COVID-19 pandemic.[20]

## NEW RECRUITMENT AND DATA COLLECTION PLANS

In 2019, funding was obtained from the Wellcome Trust to expand the GS cohort using remote data collection and extended eligibility to younger individuals (12+ years). Because of the COVID-19 outbreak in 2020, field studies other than those directly relating to the pandemic were paused. Active recruitment of new volunteers to join GS started in May 2022. Original GS cohort members have been contacted with the option to move

**Table 4** Generation Scotland genotyping and imputation summary for three imputation panels

|  | 1000G | HRC | TOPMed |
|---|---|---|---|
| Reference panel version | 1000G Phase 1 V.3 | HRC V.r1-1 | TOPMed Freeze V.5 |
| Imputation software | IMPUTE V.2 | PBWT | Minimac V.4 |
| Build | GRCh37 | GRCh37 | GRCh38 |
| Number of SNPs (post QC) | 9 438 897 | 24 161 581 | 64 616 987 |
| Indels | ✘ | ✘ | ✔ |

1000G, 1000 Genomes; HRC, Haplotype Reference Consortium; TOPMed, Trans-Omics for Precision Medicine.

**Table 5** Summary of recontact studies and the number of participating GS volunteers

| Recontact study | | Study dates | GS participants eligible for recontact | GS participants (%responded) |
|---|---|---|---|---|
| DOLORisk | Baseline questionnaire | May 2016 to December 2016 | 20 221* | 7238 (35.8%) |
| | Follow-up survey | June 2018 to June 2019 | 6657† | 5292 (79.5%) |
| STRADL | Remote questionnaire | 2015–2017 | 21 525‡ | 9905 (46.0%) |
| | Face-to-face clinic visit | | 9618§ | 1189 (12.4%) |
| COVIDLife | COVIDLife1 survey | April 2020 to June 2021 | 22 796¶ | 4968 (21.8%) |
| | Rural COVIDLife survey | October 2020 to November 2020 | 1559** | 712 (45.7%) |
| EPAD | Alzheimer's dementia | 2016–2018 | 3779†† | 53 (1.4%) |
| HAGIS | COVID-19 impact survey | 2021–2022 | 14 891 | 2826 (19.0%) |

Study eligibility criteria.
The number of eligible GS participants invited to each study varies due to study criteria. All eligible GS participants consented to recontact.
*Known email or postal address.
†Participated in the baseline DOLORisk survey and consented to recontact for the follow-up survey.
‡Living in Scotland and had a valid Community Health Index number.
§Indicated willingness to undergo face-to-face assessment in the remote STRADL questionnaire.
¶Resident in Scotland, known email address or postal address.
**Living in rural Scotland.
††Aged 50+, no diagnosis of dementia or medical/psychiatric disorders.
Aged 50+, living in Scotland.
EPAD, European Prevention of Alzheimer's Dementia Consortium; GS, Generation Scotland; HAGIS, Healthy AGeing in Scotland; STRADL, Stratifying Resilience and Depression Longitudinally.

online to complete new questionnaires and invite friends and family members to join the next phase of the study (snowball recruitment). Other recruitment methods to date have included: email invitations to Scotland-based participants of the COVIDLife study, news coverage (TV segments, radio, newspaper and online news articles), a paid TV advertisement and social media advertising.

NGS aims to recruit 20 000 new participants and will use established methods for linkage to routine NHS data to create a larger, richer, longitudinal resource. Anyone living in Scotland aged 12 years and over is eligible to join; those aged 12–15 years require parental confirmation of their capacity to consent. Participants sign-up on our online portal, complete study consents and a baseline questionnaire to collect lifestyle measures and medical history.

Saliva samples are being collected by post for genotyping of new participants. At the time of writing, over 10 000 new participants have been recruited, adding to the 2006–2011 cohort recruits.

Adolescence and early adulthood are critical periods in the development of mental and physical health.[21] The extension to younger individuals, along with potentially other family members, will make the cohort a valuable resource for research into genetic and environmental determinants of health among adolescents and young adults. There are few comparable genetic cohorts using routine data linkage in young people. Early approved studies are planned to focus on mental health, sleep and loneliness in this age group.

New questionnaires will be regularly added to the online portal to enable ongoing engagement with participants and collect enhanced data such as cognitive testing. Researchers will be able to submit approved research questions for prospective data collections. Through a broad range of recruitment strategies, participant involvement and engagement and the use of remote data collection, we hope to improve geographic coverage and sociodemographic diversity across Scotland, aiming to engage groups typically under-represented in large-scale studies. Completion of the expansion phase, combined with the original GS participants, should create an overall cohort of over 40 000 individuals across Scotland with rich genetic and phenotypic data.

### Participant and patient involvement

A key component of the GS:SFHS was to conduct a public consultation programme, which was used to ask the public their thoughts on genetics in healthcare and research and use this to develop principles of participation and data access.[22 23] Regular newsletters are distributed to participants to provide updates on the latest cohort information and recent findings. Patient and public involvement and engagement is being developed within the new NGS cohort recruitment. A survey receiving 1000 responses invited participants to become GS ambassadors in their

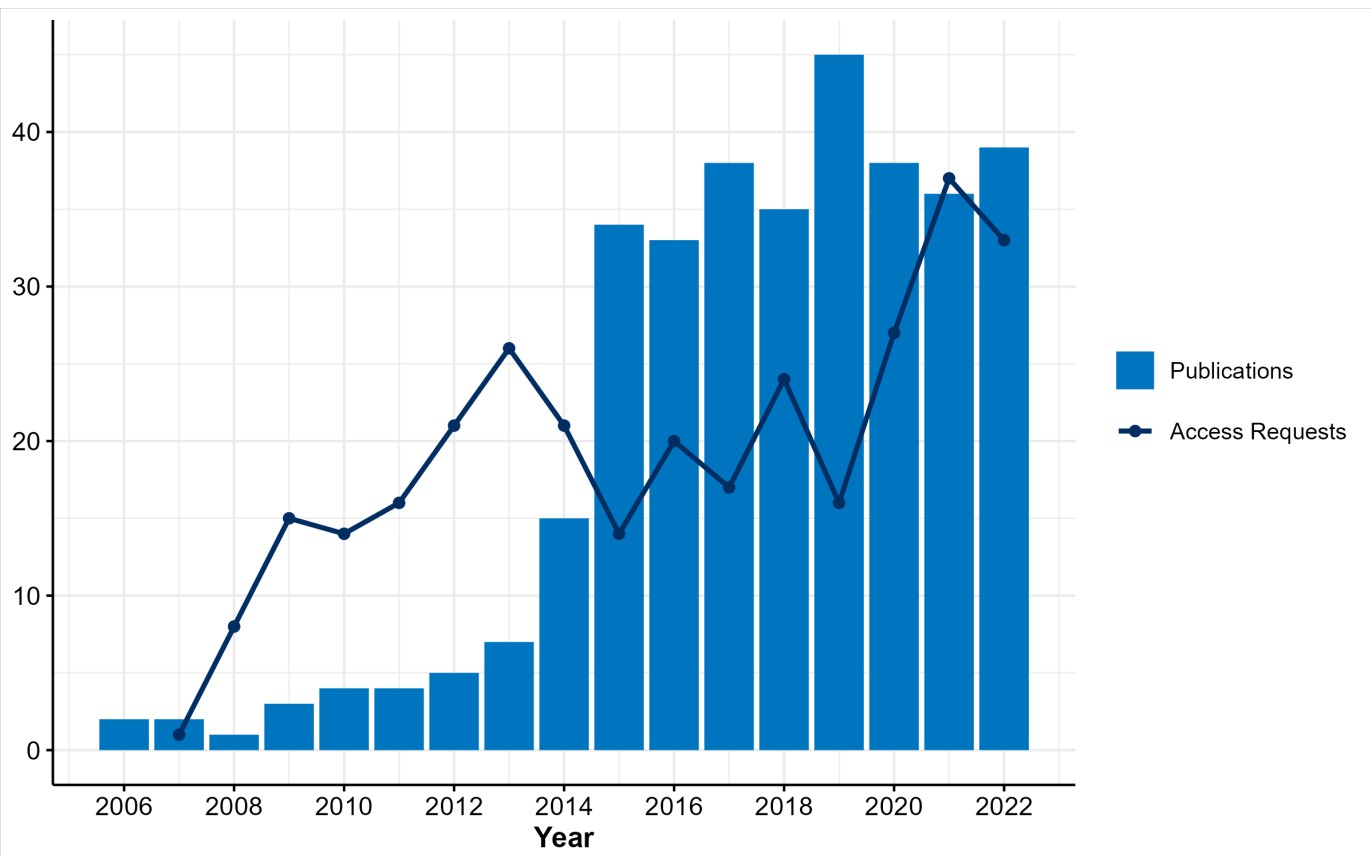

**Figure 3** Yearly number of access requests and publications using Generation Scotland data over time.

local area, take part in focus groups and test new questionnaires. These volunteers have already helped with survey testing and provided feedback on recruitment materials. Development of a Young Persons Advisory Group has helped direct teen recruitment activities and will shape future GS health research with teenagers themselves.

## Findings to date

The GS cohort has facilitated research contributions to a wide range of health conditions and scientific areas including ageing, cancer, cardiovascular disease, mental health and the role of DNAm in understanding and predicting disease. Over 350 papers have been published using GS data (figure 3). Online supplemental appendix F lists the 50 most cited papers using GS data, and a full and growing publication list can be found on the GS website (https://www.ed.ac.uk/generation-scotland/what-found/publications). Examples of some key contributions are summarised below.

Welsh *et al* used residual blood samples from GS participants (n=19 501) to assay cardiac troponin T (cTnT) and cardiac troponin I (cTnI), proteins essential for heart contraction, and to investigate their association with cardiovascular outcomes.[24] The research team identified deaths or hospitalisations of interest using Scotland's morbidity records and deaths data from GS recruitment to September 2017. They found that cTnT and cTnI were both associated with heart failure and cardiovascular disease death. Individuals with high levels of cTnT were

more likely to suffer from heart disease, stroke or other heart conditions. Troponin level testing is inexpensive, and this study demonstrated the potential benefit of testing for future health screening.

Given its size, the GS DNAm resource is well placed to serve as a training dataset for the development of risk predictors. Cheng *et al* used GS methylation data to develop and validate a model for 10-year risk prediction of type 2 diabetes.[25] They combined standard risk prediction information such as age, sex, body mass index and family history of the disease with DNAm data, which improved prediction for likelihood of developing diabetes. The results were tested using a hypothetical screening scenario of 10 000 people, which correctly classified an additional 449 individuals using methylation data compared with traditional risk factors alone.

Green *et al* investigated the aetiology of MDD among individuals from the STRADL recontact study.[26] They reported the associations of serological and methylomic signatures of C reactive protein (considered to represent acute and chronic measures of inflammation, respectively) with depression status/symptoms and structural neuroimaging phenotypes. The study provided evidence for the involvement of peripheral inflammation in brain morphology and depression symptoms and demonstrated the combined use of survey, neuroimaging, serological and methylation data from the GS cohort.

GS facilitated a pilot study to investigate the use of newborn blood spots in longitudinal research. Heel prick blood spots are used routinely to test for treatable neonatal metabolic conditions and have been retained in Scotland for all children born since 1965. Researchers showed that archival blood spots contain enough information to link to the volunteer health records, and samples were of sufficient quality to generate biologically meaningful results.[27] For example, epigenetic signatures of perinatal maternal smoking status could be identified. This pilot study confirmed the feasibility of the use of these archived newborn blood spots in a population-level retrospective birth cohort study. It has the potential to scale to a linked collection of 3 million archived blood spots across Scotland, making it one of only two such resources available worldwide.[28] Future work is dependent on a Scottish government-led public consultation to review the current pause on research access.[29]

## Strengths and limitations

Important strengths of GS are the breadth of demographic, lifestyle and health factors, and inclusion of participants from a wide range of sociodemographic backgrounds. The cohort is rich in genetic and linkage data. Scotland is ideally suited to a longitudinal cohort study given its comparatively static and stable population and relatively high prevalence of common conditions and adverse lifestyle risk factors.[1 2] The family-based recruitment approach delivers increased kinship among participants and pedigree mapping enables measurement of heritability and familial aggregation of traits.

Linkage to a variety of routine NHS datasets creates a wealth of research opportunities, while participants' consent for future recontact studies provides potential for additional data collections. Using linkage to gather longitudinal data makes the cohort more robust to attrition as passive linkage allows us to link to new data even if a participant does not take part in future data collections such as recontact studies. Planned linkages to routine NHS medical images, radiology reports and administrative data, such as education, income and benefits, will provide uniquely rich information about the participants and its relationship with future health and well-being, further enhancing the research potential of the cohort.

There are some limitations of the GS cohort. The cohort is relatively small, by contemporary standards for population-based cohorts, which can limit the statistical power to address some research questions definitively (eg, to study rare diseases or small effect sizes). However, this issue can often be addressed through joint analyses with other population-based cohorts and participation in genetic data consortia, which GS actively contributes to. The current expansion of the cohort will also help to address this limitation. Many phenotypes are assessed using self-reported measures which may be subject to recall or response bias. These potential biases are minimised in GS by using validated questionnaires applied widely in research and confirmation of outcomes through

linkage to medical records. Compared with the Scottish population, individuals in the cohort are generally older, more likely to be female and less socially deprived. This may limit the power of research studies to pick up relationships with health outcomes and factors such as education/deprivation at the lowest ends of the scale. However, it is hoped that increased diversity of the cohort will be achieved with the current expansion to reach a total cohort size of over 40 000 individuals in Scotland. GS aims to be the UK's largest multigenerational longitudinal life-course study of genetic, epigenetic, clinical, lifestyle and environmental health determinants.

## Data access and collaborations

Researchers can submit proposals to access GS data and samples through our website (https://www.ed.ac.uk/generation-scotland/for-researchers). This also includes data from recontact studies which can be accessed through a single application to GS. Research proposals are subject to review by the GS access process, under the guidance of the scientific steering committee, based on criteria set out in the management, access and publications policy. We welcome proposals for data and sample access and for prospective data collections using the NGS online portal. Further information about the cohort, details of the application process and conditions for access is available at the study website.

GS also collaborates with—and makes its data and/or metadata available via—the Dementias Platform UK (DPUK), UK Longitudinal Linkage Collaboration (UK LLC), CLOSER, BC Platforms and Health Data Research UK (HDR UK) Innovation Gateway. Access requests can be made through DPUK and UK LLC using the standard GS access process as well as directly to GS. All applications via these platforms are reviewed by the GS access process. Study metadata is available through CLOSER Discovery, BC Platforms and HDR UK.

GS genetic data have contributed to large-scale consortia including Cohorts for Heart and Aging Research in Genomic Epidemiology,[30] Chronic Kidney Disease Genetics,[31] Genetic Investigation of ANthropometric Traits,[32] SpiroMeta,[33] Global Biobank Meta-analysis Initiative,[34] COVID-19 Host Genetics Initiative,[35] Global Lipids Genetics Consortium[36] and The Psychiatric Genomics Consortium.[37]

**Author affiliations**
[1]Centre for Genomic and Experimental Medicine, Institute of Genetics and Cancer, The University of Edinburgh, Edinburgh, UK
[2]Centre for Medical Informatics, Institute of Population Health Sciences and Informatics, The University of Edinburgh Usher, Edinburgh, UK
[3]Division of Psychology, School of Humanities, Social Sciences and Law, University of Dundee, Dundee, UK
[4]Department of Psychology, The University of Edinburgh, Edinburgh, UK
[5]Division of Psychiatry, The University of Edinburgh, Edinburgh, UK
[6]Institute of Population Health Sciences and Informatics, The University of Edinburgh Usher, Edinburgh, UK

**Acknowledgements** The authors are grateful to all the people and their families who have taken part in GS to date, the general practitioners and the Scottish School of Primary Care for their help in recruiting them, the GS scientific steering committee and the whole GS team, which includes interviewers, computer and laboratory technicians, clerical workers, research scientists, volunteers, managers, receptionists, healthcare assistants and nurses.

**Contributors** The concept and design of Generation Scotland was developed by CH, REM, AMM, DJP, CLMS and HCW who obtained all funding, necessary ethical approvals and acted as principal investigators and chief scientist (HCW). Acquisition and curation of data relating to Generation Scotland were conducted by AC, CF-R, RF, DM, HM and AR. SR led patient and public involvement and engagement activities including development of a Young Persons Advisory Group. HM drafted the manuscript. All authors critically revised the manuscript for important intellectual content and approved the final version for publication. Guarantor, CH.

**Funding** GS:SFHS was funded by a grant from the Chief Scientist Office of the Scottish Government Health Directorates (CZD/16/6) and the Scottish Funding Council (HR03006). Genotyping of the GS:SFHS samples was carried out by the Genetics Core Laboratory at the Edinburgh Clinical Research Facility, University of Edinburgh, Scotland, and was funded by the Medical Research Council UK and the Wellcome Trust (Wellcome Trust Strategic Award 'STratifying Resilience and Depression Longitudinally' Reference 104036/Z/14/Z). NGS is funded by the Wellcome Trust (216767/Z/19/Z). DNAm analysis was funded by the Wellcome Trust (Reference 220857/Z/20/Z).

**Competing interests** REM is a scientific advisor to Optima Partners and the Epigenetic Clock Development Foundation. DM is a part-time employee of Optima Partners.

**Patient and public involvement** Patients and/or the public were involved in the design, or conduct, or reporting, or dissemination plans of this research. Refer to the Methods section for further details.

**Patient consent for publication** Consent obtained directly from patient(s).

**Ethics approval** This study involves human participants. All components of Generation Scotland received ethical approval from the NHS Tayside Committee on Medical Research Ethics (REC Reference Number: 05/S1401/89). Generation Scotland has also been granted Research Tissue Bank status by the East of Scotland Research Ethics Service (REC Reference Number: 20-ES-0021), providing ethical approval for a wide range of uses within medical research. Written informed consent was obtained from all participants in GS:SFHS. NGS participants gave consent online. All participants aged 16 years or over provide their own informed consent whilst those aged 12–15 years require parental confirmation of their capacity to consent. Participants gave informed consent to participate in the study before taking part.

**Provenance and peer review** Not commissioned; externally peer reviewed.

**Data availability statement** Data are available on reasonable request. Data are available on reasonable request. Researchers may request access to Generation Scotland data through our website (https://www.ed.ac.uk/generation-scotland/for-researchers).

**ORCID iDs**
Hannah Milbourn http://orcid.org/0000-0003-4479-8635
Daniel McCartney http://orcid.org/0000-0003-3242-0360
Riccardo E Marioni http://orcid.org/0000-0003-4430-4260
Andrew M McIntosh http://orcid.org/0000-0002-0198-4588
Heather C Whalley http://orcid.org/0000-0002-4505-8869

Cathie Sudlow http://orcid.org/0000-0002-7725-7520

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
