## [Reviewer comments · BMJ Open]

ARTICLE DETAILS

TITLE (PROVISIONAL)	Cohort Profile: Generation Scotland – an update on Scotland’s longitudinal family health study
AUTHORS	Milbourn, Hannah; McCartney, Daniel; Richmond, Anne; Campbell, Archie; Flaig, Robin; Robertson, Sarah; Fawns-Ritchie, Chloe; Hayward, Caroline; Marioni, Riccardo; McIntosh, Andrew; Porteous, David; Whalley, Heather; Sudlow, Cathie

VERSION 1 – REVIEW

REVIEWER	Al Anouti, Fatme Zayed University, Health Sciences
REVIEW RETURNED	11-Feb-2024

GENERAL COMMENTS	This is a cohort Profile describing the launch of a larger, richer, longitudinal resource, “Next Generation Scotland” (NGS) launched in May 2022 to expand the existing cohort by a target of 20,000 additional volunteers, now including anyone aged 12 years upwards. New participants complete online consent and questionnaires and provide postal saliva samples for DNA, from which genotype and salivary DNA methylation array data will be generated. The extension to younger individuals, along with potentially other family members, will make the cohort a valuable resource for research into genetic and environmental determinants of health among adolescents and young adults. The work is impressive and will provide insightful information no doubt. There are few minor comments listed below to consider. Longitudinal Data linkage Table 2 shows the range of datasets linked to the GS cohort, their periods of coverage and the numbers of participants with linked data available for each. Additional details are provided in Appendix A. Beyond linkages to hospital episodes, primary care, cancer and death registries and community electronic prescribing, GS has linkage to a range of other datasets, for example routine laboratory tests, dental data (from the Management Information & Dental Accounting System, MIDAS) and the Scottish Drug Misuse Database (SDMD), offering unique phenotype information distinct from other population-based cohort research resources. How are MIDAS and SDMS datasets are linked to GS, this needs clarification? Re-contact studies The Stratifying Resilience and Depression Longitudinally (STRADL) sub-study built on the GS cohort to subtype major
--

	depressive disorder (MDD) based on its aetiology, using detailed clinical, cognitive, and brain imaging assessments There is something missing in this sentence Cohort morbidities HbA1c level above 6.5% was classified as diabetic (28% of cases captured) [6]. This should be 6.5 not 6.5%
--	---

REVIEWER	Griep, Rosane Haerter Instituto Oswaldo Cruz
REVIEW RETURNED	18-Mar-2024

GENERAL COMMENTS	I am grateful for the opportunity to review this article, which presents an interesting family-based cohort study called Generation Scotland. I have few comments, overall the study is well described and the study presented is powerful for future studies in the area of genetics, lifestyle and environmental determinants of physical and mental health. In the abstract, I suggest including the age of the participants, both in the sample of adults and families and percentage by gender. Strengths and limitations highlight: I am not clear that the main limitation is the use of self-reported questionnaires. For various physical or mental health situations, the participants' self-perception is fundamental information. I consider limitations to be aspects such as a non-representative sample (I understood that they were samples of volunteers) of the population, loss of follow-up or loss of information/adherence to planned data collection. Line 45 to 47: what are the implications of a sample with a higher level of education and lower deprivation index when compared to the Scottish population? I did not find figure 1 and figure 2 in the articles.
--

VERSION 1 – AUTHOR RESPONSE

Reviewer #1

“Table 2 shows the range of datasets linked to the GS cohort, their periods of coverage and the numbers of participants with linked data available for each. Additional details are provided in Appendix A. Beyond linkages to hospital episodes, primary care, cancer and death registries and community electronic prescribing, GS has linkage to a range of other datasets, for example routine laboratory tests, dental data (from the Management Information & Dental Accounting System, MIDAS) and the Scottish Drug Misuse Database (SDMD), offering unique phenotype information distinct from other population-based cohort research resources. How are MIDAS and SDMS datasets are linked to GS, this needs clarification?”

Author response:

Thank you for your comment. As with the other datasets, these datasets are linked via the community health index (CHI). We have clarified the text accordingly:

Longitudinal data linkage, page 9, line 13:

Beyond linkages to hospital episodes, primary care, cancer and death registries and community electronic prescribing, GS has linkage to a range of other datasets via participants' CHI numbers, including routine laboratory tests, dental data (from the Management Information & Dental Accounting System, MIDAS) and the Scottish Drug Misuse Database (SDMD), offering unique phenotype information distinct from other population-based cohort research resources.

“The Stratifying Resilience and Depression Longitudinally (STRADL) sub-study built on the GS cohort to subtype major depressive disorder (MDD) based on its aetiology, using detailed clinical, cognitive, and brain imaging assessments. There is something missing in this sentence.”

Author response: We apologise for the confusion. To improve the clarity of this sentence the paragraph has been revised as follows:

Re-contact studies, page 9, line 16:

The Stratifying Resilience and Depression Longitudinally (STRADL) sub-study recruited from the existing GS cohort to subtype major depressive disorder (MDD) based on its aetiology, using detailed clinical, cognitive, and brain imaging assessments.

“HbA1c level above 6.5% was classified as diabetic (28% of cases captured) [6]. This should be 6.5 not 6.5%.”

Author response: We have double checked this figure and we can verify that the original units were correct. The hemoglobin A1c (HbA1c) test referred to in this sentence measures the proportion of glycated haemoglobin in the blood which is measured as a percentage. We have updated this sentence to clarify the units of measurement and have added the reference for this cut off:

Cohort morbidities, page 11, line 16:

Individuals with percentage glycated haemoglobin in blood (HbA1c levels) above 6.5 (48mmol/mol) was classified as diabetic (28% of cases captured) [6].

Reviewer #2

“I suggest including the age of the participants, both in the sample of adults and families and percentage by gender.”

Author response: We thank the reviewer for this suggestion. The mean age and sex distribution has been added to the abstract to make clear the broad demographics of the cohort.

Abstract, page 4, line 6:

Participants: *A total of 24,084 adult participants, including 5,501 families, were recruited between 2006 and 2011. Within the cohort 59% are female, with an average age at recruitment of 49 years.*

“Strengths and limitations highlight: I am not clear that the main limitation is the use of self-reported questionnaires. For various physical or mental health situations, the participants' self-perception is fundamental information. I consider limitations to be aspects such as a non-representative sample (I understood that they were samples of volunteers) of the population, loss of follow-up or loss of information/adherence to planned data collection.”

Author response: This is an interesting point raised by the reviewer. When listing self-report as a limitation in this section we intended to highlight issues such as recall or response bias (for example: alcohol consumed in last seven days; cigarettes smoked per day). These biases could lead to inaccurate estimates of association to health outcomes. The text has been updated to reflect that these biases are not inherent.

Strengths and limitations, page 18, line 22:

Many phenotypes are assessed using self-reported measures which may be subject to recall or response bias. These potential biases are minimised in GS by using validated questionnaires applied widely in research and confirmation of outcomes through linkage to medical records.

The strengths and limitations bullet points section has also been updated to reflect that the cohort size is the main limitation of the cohort rather than use of self-reported measures.

Strengths and limitations, page 5, line 10:

- *The cohort is relatively small, by contemporary standards for population-based cohorts. However, this issue can often be addressed through joint analyses with other population-based cohorts and participation in genetic data consortia.*

The reviewer also poses that loss to follow-up could be a limitation of the cohort. Although this is a common limitation for longitudinal studies this is less a limitation for Generation Scotland since our cohort relies extensively on data linkage for the longitudinal aspect of the cohort. We have clarified this as a strength of the cohort in the text.

Strengths and limitations, page 18, line 11:

Using linkage to gather longitudinal data makes the cohort more robust to attrition as passive linkage allows us to link to new data even if a participant does not take part in future data collections such as re-contact studies.

“Line 45 to 47: what are the implications of a sample with a higher level of education and lower deprivation index when compared to the Scottish population?”

Author response: In response to this comment additional comment has been added to this section:

Strengths and limitations, page 18, line 26:

This may limit the power of research studies to pick up relationships with health outcomes and factors such as education/deprivation at the lowest ends of the scale.

“I did not find figure 1 and figure 2 in the articles.”

Author response: Figures were uploaded with the manuscript during the submission process. These may have been lost between submission and review. We have re-attached figures in the response for review, and apologise for any oversight.

- FAO editor to highlight figures.

VERSION 2 – REVIEW

REVIEWER	Griep, Rosane Haerter Instituto Oswaldo Cruz
REVIEW RETURNED	27-May-2024
GENERAL COMMENTS	Thanks for improvement of the article. Hope you have success with the article and Generation Scotland study

VERSION 2 – AUTHOR RESPONSE